# Can We Use Gene-Editing to Induce Apomixis in Sexual Plants?

**DOI:** 10.3390/genes11070781

**Published:** 2020-07-12

**Authors:** Armin Scheben, Diego Hojsgaard

**Affiliations:** 1Simons Center for Quantitative Biology, Cold Spring Harbor Laboratory, Cold Spring Harbor, NY 11724, USA; scheben@cshl.edu; 2Department of Systematics, Biodiversity and Evolution of Plants, Albrecht-von-Haller Institute for Plant Sciences, University of Goettingen, Untere Karspuele 2, 37073 Goettingen, Germany

**Keywords:** apomeiosis, character segregation, crop biotechnology, heterosis, meiosis, molecular breeding, recombination

## Abstract

Apomixis, the asexual formation of seeds, is a potentially valuable agricultural trait. Inducing apomixis in sexual crop plants would, for example, allow breeders to fix heterosis in hybrid seeds and rapidly generate doubled haploid crop lines. Molecular models explain the emergence of functional apomixis, i.e., apomeiosis + parthenogenesis + endosperm development, as resulting from a combination of genetic or epigenetic changes that coordinate altered molecular and developmental steps to form clonal seeds. Apomixis-like features and synthetic clonal seeds have been induced with limited success in the sexual plants rice and maize by using gene editing to mutate genes related to meiosis and fertility or via egg-cell specific expression of embryogenesis genes. Inducing functional apomixis and increasing the penetrance of apomictic seed production will be important for commercial deployment of the trait. Optimizing the induction of apomixis with gene editing strategies that use known targets as well as identifying alternative targets will be possible by better understanding natural genetic variation in apomictic species. With the growing availability of genomic data and precise gene editing tools, we are making substantial progress towards engineering apomictic crops.

## 1. Introduction

The formation of clonal offspring through parthenogenesis is a well-known feature of many phylogenetically distant organisms [1,2,3,4,5], with shared developmental features at least in flowering plants and vertebrate animals [6]. Yet, in most cases, details of the genetic basis and molecular coordination of parthenogenetic development are lacking. 

In plants, parthenogenesis holds a gargantuan economic dimension because of its probable impact on breeding. Sexuality imposes critical disadvantages to genetic improvement, making the exploitation of hybrid vigor and high yield only accessible in the short time frame of a single F_1_ generation. This raises the need for costly and laborious methods to identify parental lines with optimal ’combining ability,’ i.e., inbred lines which complement each other for desired traits in the hybrid [7]. In F_1_ hybrid breeding procedures, such a testing of inbred lines for their combining ability is the most limiting factor [7]. Since clonal seed embryos can be created through parthenogenesis (i.e., embryo development from an unfertilized egg cell), engineering parthenogenesis in sexual crops has long been a goal for many researchers around the world. Introducing parthenogenesis into plant breeding would both simplify the selection of parental lines and extend over time the exploitation of the desired F_1_ hybrid (expectedly through many generations) breaking the breeding loop in traditional schemes.

However, unlike in animals, in plants the formation of parthenogenetic individuals involves a higher level of complexity as seeds are formed literally by two individuals, the embryo per se and the endosperm, and a second individual that has evolved to have an acquiescent temporary role nourishing the embryo [8]. Thus, the formation of a seed involves twice the number of gametes, a double fertilization step to initiate the embryo and the endosperm developments and more complex molecular controls to produce a functional, viable seed. First, a megaspore mother cell (MMC) in the ovule goes through a meiotic division and produces four (sometimes three) reduced (haploid or *n*) megaspores. Only one megaspore grows and differentiates by mitosis into a multicellular female gametophyte carrying two gametes with different ploidy, a haploid egg cell (*n*) and a diploid central cell (two haploid nuclei, *n* + *n*). Both female gametes are fertilized by haploid sperms delivered by the pollen tube to produce a diploid zygote and a triploid primary endosperm, which develop into the embryo and endosperm tissues of the sexual seed, respectively. Hence, along such reproductive development in ovules of diverse species of flowering plants, there are a few critical developmental steps which can be divided into defined reproductive modules or developmental programs, each having specific molecular controls and functional roles during seed formation (Figure 1). 

Some plants can alter these controls along different reproductive steps and develop parthenogenetic embryos through apomixis. Thus, apomixis exploits natural developmental programs for the formation of parthenogenetic embryos and clonal seeds. In plants with sporophytic apomixis, the formation of the female gametophyte and the successive developmental steps are not altered, but extra somatic embryos are attached to the gametophyte usually constraining the development of the zygotic embryo.

In plants with gametophytic apomixis, the developed female gametophytes are unreduced (2*n*), produced either from the germline going through a modified meiotic division sidestepping genetic recombination (diplospory) or from a somatic cell in the ovule nucellus without going through a meiotic division (apospory). Like in sexual ovules, unreduced female gametes show different ploidy, the egg cell being diploid (2*n*) and the central cell (usually) being tetraploid (2*n* + 2*n*). However, in apomictic ovules the egg cell develops by parthenogenesis into an embryo genetically equal to the mother plant, and the central cell is often fertilized by one (rarely more) haploid (*n*) sperm(s) to produce the endosperm tissue of the apomictic seed.

Thus, the so-called ‘*elements of apomixis*’ [2] are alterations in the functional output of successive sexual reproductive modules (i.e., meiosis, gametogenesis, and fertilization steps) that do not necessarily interfere with each other. However, since the molecular basis of apomixis is unknown, some factual ambiguities in the developmental features and occurrence of these altered reproductive modules cannot yet be explained. First, even though apomictic ovules show global gene de-regulation (on genes affecting varied functions) and heterochronic developments compared to sexual ones (e.g., [9]), there seem to be concerted regulatory changes acting in coordination to achieve the goal of forming a seed. Second, even when developmental programs (in terms of alleles and gene networks) are expectedly the same between diploids and polyploids (at least in autopolyploids), apomixis is transgenerationally stable [10], penetrant, and highly expressed only in polyploids. Third, while apomixis in nature is dominant over sexuality (though it often shows segregation distortion; [11]), apomixis-like mutants in sexual plants have recessive phenotypes.

In the last 30 years, different plant mutants affecting specific developmental steps in those reproductive modules have been described [12,13]. Many of these reproductive mutants display phenotypes resembling alterations to the sexual developmental programs observed in natural apomictic plants. More recently, the arrival of gene editing methods has sped up the use of several of these plant reproduction mutants individually or in combination to simultaneously alter the normal output of each developmental program and reproductive module within plant ovules, with the goal of obtaining plants simulating natural apomixis. Although these studies successfully produced multiple concurrent mutants, they had relatively low success on attaining a synthetic apomictic plant exhibiting high expressivity. Therefore, the question remains whether apomixis per se can be induced in sexual plants. Here, we address this question by discussing the genetic control of apomixis and how gene editing approaches can be used to induce this complex trait.

## 2. Precise Gene-Editing of Complex Traits

Although most genetic trait modifications in crops still rely on gene knockouts, in recent years the gene editing toolbox has expanded to facilitate many other modifications of functional sequences [14,15]. Gene knockouts nevertheless remain a powerful approach and have induced several apomixis-related traits in plants (see details in Section 5 on *de novo* apomixis). Complex traits result from the interaction among multiple genes and their environment. Induction of a complex trait such as apomixis will likely require a combination of different types of gene editing. Up to now, attempts to induce synthetic apomixis are limited by low expressivity of the trait in modified sexual plants. To efficiently introduce the apomictic pathway into sexual plants, researchers can use modification of gene expression, precise base editing and trait optimization using allelic series.

### 2.1. Modulating Gene Expression

Gene expression can be modulated by editing regulatory machinery such as promotors and transcription factors. Particularly in polyploids, an additional strategy to reduce gene expression can also be the knockout of redundant paralogs. Editing approaches to increase expression include promoter knockin and upstream open reading frame editing [16]. For example, a promoter swap targeting the *ARGOS8* gene in maize induced overexpression of the gene in multiple tissues and during different developmental stages [17]. This overexpression was associated with a gain-of-function that led to increased drought tolerance.

Complementing nucleotide-level editing to modify gene expression, catalytically inactive ‘dead’ Cas (dCas) proteins can also act as a recruiting platform for repressors/activators [18] and epigenetic modifiers. In recent years, several approaches have been optimized in plant cells. By joining transcriptional activators such as VP64 and TV to dCas9, expression of a targeted luciferase reporter gene could be increased 2.4-fold to 215-fold, respectively [19]. A combination of VP64 with a modified guide RNA can increase the effectiveness of this activator and a system based on transcription activator-like effector nucleases (TALENs) may be even more effective [20]. The level of activation possible using these systems can differ by gene and further optimization may be achieved by testing combinations of single and multiple activators with a targeting system. 

A further approach to modulate expression is to make modifications at the epigenome level. Epigenomic features such as DNA methylation and histone modification play important roles in gene regulation. By fusing dCas9 with histone acetyltransferases, methyltransferases or demethylases, gene expression can be activated or repressed by changing epigenomic states. Site-specific methylation and demethylation using dCas9 together with the SunTag system has been demonstrated to be effective in *Arabidopsis* [21,22,23]. The SunTag system links the Cas molecule with an GCN4 peptide array that can recruit specific antibodies with attached epigenetic modifiers or transcription activators/repressors. This approach could be used to down- or upregulate the expression of target reproduction genes. However, off-target effects can occur, including hypermethylation in large regions of the genome [21]. The specificity of epigenome editing therefore needs to be increased, particularly when stably modifying complex pathways in crops for commercial use. A further challenge when using this epigenome editing approach in crops is that the CRISPR/Cas transgenes must remain in the plant, triggering governmental regulation in many markets. Although achieving high specificity and efficiency remain challenging, the modulation of gene expression on the genomic and epigenomic level can help reprogram complex reproductive gene networks.

### 2.2. Targeting Gene Sequence with Base Editing and Prime Editing

Gain-of-function mutations that modify how a protein works can be induced by point mutations, precise indels in functional domains, or gene replacement. This type of mutation can be induced in plants with moderate efficiency (<10%) using gene targeting that relies on double strand breaks and DNA template [24]. However, this gene targeting approach mostly generates imprecise indels. As an alternative, the more precise base editing can be used to generate point mutations, without requiring DNA template or induction of double strand breaks [25]. The critical component of base editing is a fusion of a cytidine deaminase [26] or adenosine deaminase [27] to a Cas9 nickase (nCas9), which only breaks a single DNA strand, or dCas9. The fusion proteins enable four types of transition point mutations (C→T, G→A, A→G, and T→C). A drawback of base editors is that they can generate off-target point mutations within the deamination window spanning several bases. In rice, cytosine base editors have also been found to introduce off-target mutations outside of the target region [28]. Finally, a broader limitation of base editing is that it cannot be used to perform all possible base-to-base conversion or to introduce indels.

A substantial advance in gene targeting was the recent introduction of prime editing [29], which has increased gene targeting efficiency in plants to >20% [30]. The first component of prime editing is a fusion protein consisting of an nCas9 and a reverse transcriptase. The second component is a prime editing guide RNA that includes both a guide that is homologous to the genomic target and an RNA template with the desired edit. The nCas9 nicks the target location and the RNA edit is reverse transcribed to DNA and written into the genome. Prime editing is now the most versatile and efficient gene targeting approach in most situations, enabling the introduction of all 12 base-to-base conversions as well as indels [29]. Transition point mutations can, however, still be introduced more efficiently using base editors [30]. Prime editing is also less limited by the availability of protospacer adjacent motif (PAM) sequences than base editing or other editing methods [29], making it possible to target more regions of the genome. Prime editing efficiency, however, decreases with the length of the introduced indel and remains restricted to small indels <20 nt [30]. Importantly, the relative efficiencies of different gene targeting approaches vary by target site [30], suggesting that case-by-case optimization for each target will be important when tackling complex traits such as apomixis.

### 2.3. Trait Optimization with Allelic Series

When an edit that affects the target trait has been identified, the trait can be optimized by inducing a range of allelic variants and combinations of these variants [31]. Such an allelic series can identify functional variants with diverse impacts on a trait. For example, a series of cis-regulatory alleles of the *CLAVATA3* gene in tomato generated step-wise variation in fruit size [32]. In this case different expression levels of the target gene were associated with the trait variation, but different allelic variants can also impact gene function in different ways such as loss or gain of functions [33]. Prior information on target genes and pathways, including on epistasis, protein structure, and natural variation, can help select candidate edits to use for optimization of a trait.

Multiplexed gene editing is a useful tool to accelerate the combination of different edited alleles. In plants, up to 107 simultaneous targeting events have been reported [34]. A simple case of multiplexed gene editing is the knockout of functionally redundant paralogs or members of a gene family to disrupt a function. However, guide RNAs can also be targeted to a complex combination of regulatory regions and coding sequences in less similar genes. Mixing and matching edits in this way can facilitate the manipulation of complex plant traits such as apomixis. Provided the molecular nature of apomixis is known, a certain level of confidence can be assumed that inducing *de novo* apomixis would be a technically demanding but feasible task.

## 3. The Molecular Basis of Apomixis: Three Models to Explain Empirical Data

One of the main challenges for manipulating apomixis in plant breeding is the lack of a molecular model able to suit all empirical data collected today about apomixis (see Hojsgaard [35] in this issue). In 1990, Savidan wrote a an article [36] summarizing all available information about the genetic control of apomixis and pointed out that around 95% of all data was inconclusive. Today, 30 years after Savidan’s review, we have a lot more information, particularly on the genetic regulation of sexual and apomixis-like developments (see e.g., Vijverberg et al. [37]; Ozias-Akins et al. [38]), and yet no unified hypothesis explaining the origin of apomixis exists. Three hypotheses based on different molecular mechanisms are often used to explain the observed diversity of results about asexual seed formation coming from different plant systems.

### 3.1. Apomixis is a Consequence of Developmental Asynchronies

The first hypothesis is—to some extent—related to the hybridization theory discussed by Juel [39], Murbeck [40], and Ostenfeld [41] at the beginning of the 20th century based on the observation that many *Hieracium* apomictic species were hybrids. Sexual parental types were considered to have apomictic tendencies that would reveal in the hybrids. Later, Rosenberg [42], Winge [43], and Ernst [44] added polyploidy as another prerequisite for apomixis. Of course, back then the DNA molecule was yet to be discovered, and the Mendelian theory of inheritance was just starting to be examined and accepted in a variety of organisms. 

More recently, diverse studies on ovule development in related sexual and apomictic species have exposed obvious temporal and spatial developmental asynchronies during seed formation in apomicts pointing to apomixis as a consequence of de-regulation of the sexual program, likely due to hybridization and/or polyploidization [45,46,47,48,49,50]. According to this hypothesis, apomixis in hybrids (mainly allopolyploids) arises as a consequence of the evolutionary divergence of regulatory sequences controlling sporogenesis and gametogenesis in parental species, and in polyploids (mainly autopolyploids) arises as a consequence of dosage effects and stoichiometric disbalances of macromolecular complexes. This idea has nowadays been reinforced by gene expression and transcriptomic analyses on different plant species exposing significant changes in expression levels (up- and down-regulations) of many genes in apomictic compared to sexual ovules [51,52,53,54,55,56]. Such regulatory alterations might well support weaker controls on cell fate and key developmental steps allowing, e.g., nucellar cells to acquire a gametogenesis fate (in apospory), or the primary endosperm to develop under imbalanced paternal to maternal genome ratios. However, this hypothesis cannot explain observations on genetic inheritance studies showing reproductive modules in apomicts had independent Mendelian segregation, in some cases showing the expected segregation pattern [57] and in others showing patterns distorted by secondary genetic effects including epistasis, segregation distortion, polyploidy, and lack of recombination (reviewed in [11]).

### 3.2. Apomixis is a Mutation-Based Phenomenon 

For a long time, apomixis components were assumed to have independent inheritance supported by the rare occurrence of B_III_ hybrids due to spontaneous uncoupling of reproductive modules (meiosis and fertilization) [2]. Studies focused on understanding the genetic control of apomixis used experimental crosses between sexual and apomictic plants and their segregating progeny [2,11]. Such studies usually provided inconclusive results and conflicting ideas [36], from models postulating apomixis was a delicate balance of many recessive genes [58] or the result of the action of three recessive genes [59] to models in which apomixis is controlled by a single dominant gene [60,61,62]. The introduction of different methods of molecular biology, especially in genetics and computing, facilitated the use of larger progenies as well as inheritance and genetic mapping analysis of higher complexity. Today, most apomictic species studied show Mendelian segregation of one or two dominant genes, often modulated by segregation distortion, modifiers genes, epistasis, polyploidy, aneuploidy, etc. (see details in Ozias-Akins and van Dijk [11]), although in some cases more genes had been suggested to regulate apomixis expression [63]. Thus, the overall evidence suggests apomixis is a mutation-based anomaly that involves a relatively simple genetic locus or two loci [11,64] carrying genes for apomeiosis and parthenogenesis. Even though this hypothesis is supported by diverse mutants genes showing diplosporous-like or aposporous-like and parthenogenetic phenotypes [65,66,67], it is tricky to explain an independent evolution of mutants for each apomixis component in an ancestral population. A possible explanation may lie in the characteristics of the apomixis locus. In several apomictic species showing monogenic or digenic inheritance, the apomixis locus has been identified with large non-recombinant chromosomal segments likely carrying many more genes [68,69,70,71] that are microsyntenic to chromosomal segments conserved among different species, including sexual taxa [72]. These features place the possibility of a single-event mechanism involved in the evolutionary origin of apomixis and may suggest a concerted multigenic activity in the control of reproductive modules [35]. 

### 3.3. Apomixis is an Ancient Switch, Polyphenic to Sex, and Epigentically Regulated 

More recently, Albertini et al. [73] discussed the idea that apomixis may be anciently polyphenic with sex, with both reproductive phenisms involving canalized components of complex molecular processes. According to the polyphenism viewpoint, under different environmental conditions, plants can switch on/off certain genes, change the metabolic status in ovules, and consequently, choose between an apomictic or sexual program for seed formation. Although the same genome would encode both sex and apomixis, according with this view, apomixis fails to occur in obligately sexual eukaryotes because genetic or epigenetic modifications have silenced the primitive sex-apomixis switch and/or disrupted molecular capacities for apomixis. Thus, apomixis would be an ancient character epigenetically regulated with a relictual presence in all eukaryotes [74], a view that has gained interest based on methylation analyses of apomicts and on studies in mutants affecting methylation pathways [75,76,77,78].

Each of these hypotheses suggests a different molecular frame for the molecular manipulation of sexuality and induction of apomixis in plants via *heterochrony*, *de novo* through mutations or by *restitutio* of an ancient polyphenic switch. 

## 4. Can Apomixis *Sensu Stricto* be Induced Through Gene-Editing Approaches?

As of the current state of the art, apomixis *sensu stricto* cannot be induced. Unless the specific molecular basis of apomixis is revealed, scientists will not be able to induce apomixis as we know it from natural plants by simply manipulating a few genes in a sexual plant (however, see the next section on *de novo* apomixis). 

However, framing that possibility under the different models of the regulatory control of apomixis is a good exercise to bring up points of relevance for genetic engineering. Each of the different models suggest inducing apomixis per se will be challenging. Exploiting gene editing for induction of apomixis is confronted with different obstacles depending on the type of molecular control. 

### 4.1. Apomixis Caused by Heterochronic Gene Expression 

In this case apomixis is a consequence of heterochronic gene expression derived from divergent evolution or stoichiometric disbalances of macromolecular complexes; inducing apomixis would be a formidable if not quixotic task unless a few molecular edits at sequence level are enough to mimic global regulatory changes and alter the output of specific reproductive modules.

Even in such a case, there would be several hurdles to induce apomixis-like phenotypes. For example, in hybrid apomicts formed from two putatively diverged parents, no information on sequence divergence at gene level exist for most cases. Retrieving such information from genetic data on specific apomictic individuals might be demanding as dosage of alleles (e.g., AABC, ABBC, ABCC) is difficult to obtain [79,80], and using sequence level information from parental plant materials might provide biased information depending on the apomicts age, recombination rates, and rates of molecular divergence. To sum up, it is not possible to know a priori which genes should be modified, nor the extent of the changes needed to shift molecular interactions enough to create a developmental asynchrony able to induce functional changes in the output of each reproductive module without underrunning or preventing it.

In the case of autopolyploids with higher allele similarity (e.g., A_1_A_1_A_2_A_2_), creating disbalances in stoichiometry of macromolecular complexes that are sufficient to shift reproductive pathways is hardly an option. Besides the above challenges, scientists must deal with difficulties of allelic bias, overdispersion and outliers when modelling autopolyploids [81]. Thus, lack of information about quantity and quality of allelic deviations required for an operating asynchronous development would be a barrier to inducing apomixis under this regulatory model.

In either case (alloploid or autoploid genomes), integrating genomic data and well-designed studies collecting proteomic and metabolomic data will facilitate exposing gene-protein interactions, as well as recognizing proteins and molecular complexes with relevant functions during apomixis emergence. These studies could also reveal the extent of disturbance in cellular metabolic pathways that is tolerated without triggering apoptosis.

### 4.2. Apomixis Caused by a Few Genes

In the case that apomixis is controlled by a few genes, we assume the existence of single ’master’ genes governing independent developmental programs and shifts in the functional output of each reproductive module. Specifically, one master gene for changing meiosis into an apomeiosis, one for inducing parthenogenetic development of the embryo, and one controlling the initiation/progression of endosperm development (a mechanism not yet clarified but likely regulated epigenetically). Activation of those master genes would be enough to initiate multiple concurrent changes observed at the gene level [37,38]. Thus, inducing apomixis might only require targeting those master genes (see Figure 2 and the next section on *de novo* apomixis). By choosing the right combination of sequence-level changes and regulatory modifications, a genotype could be engineered that holds the level of gene expression needed for the correct interaction between gene networks and macromolecular complexes. Moreover, this genotype would navigate changes in cell cycles and ensure a coordinated progression throughout reproductive modules and altered developmental programs to finally produce an asexual seed.

Although this may appear comprehensive from a theoretical viewpoint, it includes many drawbacks from an empirical and technical perspective. We know chromosomal regions associated with the control of apomixis present high allelic divergence, activity of transposable elements, and mutational degradation [68,72,82,83,84]. Interruptions of gene sequences located in apomixis loci suggest deregulation of apomixis-related genes may be more complex than expected and involve snRNAs and RNAi players [84,85,86,87]. Such changes on individual genes may not be easy to mimic using gene editing.

**Figure 2 genes-11-00781-f002:**
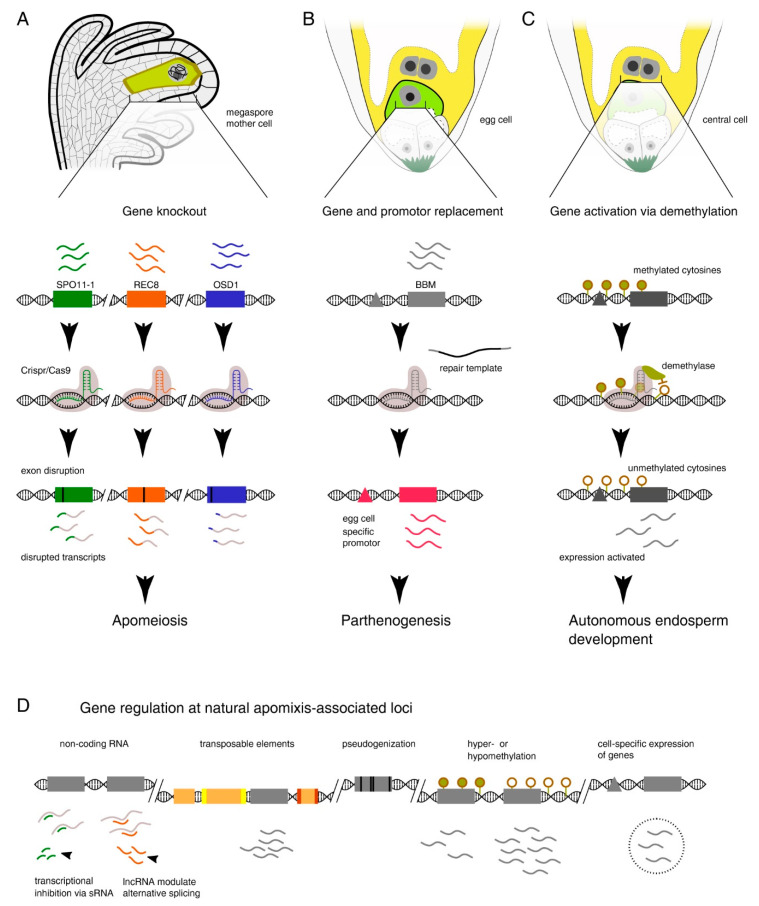
Illustration of gene editing approaches that could be used to target apomixis master genes or multiple genes affecting specific developmental steps and having a master-like effect (for simplicity, we display known gene names associated with reproductive modules and targeted in *de novo* apomixis strategies). (**A**) Gene knockout can be used to convert meiosis to mitosis, as has been shown using the Mitosis instead of Meiosis (MiMe) mutants. (**B**) Tissue-specific promotors and targeted gene replacement for a gene such as *BBM* could induce embryo development without fertilization. (**C**) Using Cas9 as a recruiting platform for demethylase would allow activation of repressed female genes, for instance to erase genomic imprinting and trigger autonomous endosperm development. (**D**) Features of observed genomic and regulatory variation at natural apomixis loci such as those known from *Pennisetum/Cenchrus* [82] and *Paspalum* [86,87,88].

Another hurdle might relate to the hemizygous condition of sequences linked to apomixis [68,69,89]. Sequences were considered hemizygous, because previous studies often did not detect an allele other than the one associated with apomixis. Hemizygosity may well be considered as an advantage for exploiting gene editing techniques, as modifying only one allele of the polyploid set for each gene would expectedly be enough to express apomixis. However, it also implies that a sexual plant may not carry the sequences required to be modified for apomixis expression. In this case, insertion of the functional alleles using an approach like homology-directed repair or prime editing may be an option to overcome the hurdle of hemizygosity. It remains difficult to insert sequences longer than several kilobases via gene editing, thus, alternative transformation approaches may be required if large sequences need to be inserted.

The *PsASGR*-*BABY BOOM*-*LIKE* gene (*BBML*), a gene involved in the induction of parthenogenesis located in an hemizygous region of the apospory-specific genomic region (ASGR) in *Pennisetum/Cenchrus*, is the only trait-associated gene isolated from an apomictic species and tested in intra- and interspecific sexual species (i.e., *Arabidopsis*, rice, maize) [67,90]. Orthologous genes present in sexual *Brassica* and *Arabidopsis* plants can induce embryogenesis [91], meaning the *BBML* family gene has a relevant function enabling embryo development (activating parthenogenesis). However, its specific molecular function and interconnection to other molecular players remains to be solved, and is likely to have a part in the observed low penetrance and its species-specific activity.

Considerable developmental variation has been observed among natural apomicts (diplosporous *Antennaria*-, *Taraxacum*-, *Ixeris*-, *Eragrostis*-types, or aposporous *Hieracium*- and *Panicum*-types of apomixis with other less frequent developmental deviations, plus pseudogamous or autonomous development of the endosperm, [2]) and several genetic mutants resembling apomixis-like phenotypes (see details in the next section). This suggest that apomixis depends on alteration of different species-specific genes controlling and/or coordinating developmental steps within reproductive modules to produce genetically balanced clonal progeny. Editing one or several genes related to each reproductive module, and generating a series of allelic variants, will be useful to induce functional changes in proteins and to modify the output of the reproductive modules. However, the impact of such changes on the coordinated development of reproductive modules within the ovule and the stable formation of apomictic seeds remain to be tested.

### 4.3. Apomixis Caused by Epigenetic Signals 

In the case apomixis is controlled by epigenetic signals which determine cellular metabolic pathways opting between sexual or apomictic developments, manipulating apomixis would be challenging. Canonical or noncanonical signaling may be needed for epigenetically channeling shifts in cellular metabolic programs. The molecular context of each metabolic pathway involved in the steps to sex or apomixis might have substantial relevance, making the induction of apomixis genotype-dependent. The cellular metabolism influences chromatin dynamics and epigenetics and thereby has functional roles in genome regulation [92]. The idea of reproductive phenisms involving anciently canalized components of complex molecular processes [73] might suit Waddington’s image of “epigenetic landscapes,” depicting valleys for stable cellular states and cell lineage specification [93]. However, quantitatively mapping epigenetic landscapes to provide predictive models of cellular differentiation and to identify optimal routes of cell fate reprogramming is not a simple task [94].

On the other hand, according to metabolic control theory, gene networks and individual metabolic pathway architecture coevolve and constrain evolutionary change through selective partitioning of metabolic fluxes into alternate channels [95]. While a *de novo* induction of apomixis implies placing altered genes in a genome not adapted to their mutated functions and hence lacking *ab initio* a buffering background, a scenario of *restitutio* of an ancient switch provisionally blocked may provide an appropriate background, delivering suitably partitioned metabolic channels. Both sexuality and apomixis are stable and penetrant in those plants in which they occur. This fit well with the hypothesis of an ancient polyphenic switch partitioning sex and apomixis into alternate channels depending upon metabolic status [73]. The absence of apomixis-like mutants displaying high expressivity might well be a consequence of the partial or incomplete partition of metabolic fluxes, and highlight the relevance of understanding protein–protein interaction networks (see discussion in the next section).

While sexuality uses regular developmental programs to form a seed, apomixis changes several regulatory mechanisms of the sexual program to do so. These mechanisms include epigenetic signals, the activity of transcription factors, non-coding RNA signaling and modulation of gene expression, protein turnover, cell-to-cell and hormonal signal transduction [9]. Having all these mechanisms canalized by an ancient switch will clearly ease the induction of apomixis, if only simple internal or external signals are required. For instance, switching between asexuality and sex in lower eukaryotes can be a response to sensed environmental signals. However, in plants such a decision would regulate whether to allocate energy into flowering, i.e., (sexual) reproduction [96]. Between lower and higher eukaryotes (such as plants), evolution has over imposed much more developmental complexity (in size and topology of gene and metabolic networks [97]) on top of the ancient switch between asexual-sexual growth (reproduction). Hence, it might be unlikely that once flowering is decided, one (or more) master-like signal(s) could channel a reproductive switch back to asexual reproduction, without being backed up with genetic redundancy controlling decisions during seed formation. Such redundancy is observed in different plant mutants [98].

Taken together, all collected data on apomixis phenotypes suggest the expression of apomixis involves permanent (genetic) or temporal (epigenetic) modification of several genes and regulatory changes [9,99]. Instead of creating developmental chaos, these modifications seem to provide a coordinated developmental flexibility between reproductive modules, and maintain high levels of expressivity in individual plants (Figure 3). Stable induction of apomixis *sensu stricto* in a sexual plant will likely require knowing most of these changes from the phylogenetically closest apomictic relative. Multiplexed gene editing would need to combine numerous edits optimized to the target genetic background.

## 5. Closing in on *De Novo* Apomixis: Making Rudimentary Changes in Reproductive Modules to Synthesize Clonal Seeds

Initiating apomixis in a sexual plant through the same molecular mechanism responsible for its occurrence in natural species is not yet a feasible task. Besides the regulatory complexity of sexual seed formation, understanding apomixis as a consequence of specific alterations in reproductive modules has allowed researchers to progress on *de novo* engineering clonal seed formation in otherwise sexual plants. A high number of genes and pseudogenes associated with apomixis or apomixis-like phenotypes in sexual and apomictic plants are enabling the manipulation of meiotic recombination [9,37,98,100,101,102,103]. Gene knockouts have induced several apomixis-related traits in plants (Table 1). Promising results were recently published based on attempts to synthesize apomixis in sexual plants by targeted modification and combination of mutants that complement key steps in reproductive modules [104,105].

### 5.1. Mimicking Sporophytic Apomixis 

The simplest road to inducing production of clonal seeds is to mimic sporophytic apomixis. Inducing an ectopic embryo within the ovule while arresting or delaying egg cell progression or zygote development in the fertilized meiotic female gametophyte may involve as few as two genes.

A simple development like this could exploit knowledge of somatic embryogenesis (e.g., *RKW* or *BBM* gene families [99,102]), and a gene (partly) arresting egg cell progression. Here, however, fertilization of the gametophyte must not be avoided, as the endosperm is crucial to the development of a viable seed. While complete arrest of zygote development will impose arrestment of the endosperm and seed failure due to embryo-endosperm signaling and communication [118,119], a temporal interruption of the development of the sexual embryo might well give advantages for somatic embryos to hoard resources from the nucellus and endosperm, as usually observed in plant exhibiting sporophytic apomixis [120].

Studies in *Citrus* had shown somatic embryogenesis is likely regulated by *CiRKD1*, a gene encoding an RWP-RK domain-containing transcription factor [121]. *RKD* genes are expressed in egg and synergid cells of different species, including sexual *Arabidopsis thaliana* and apomictic *Boechera gunnisoniana* [122,123,124,125] and likely play a role in cell fate, cell identity in absence of fertilization, and acquisition of embryogenic competence. In *Citrus*, one of the two characterized *CiRKD1* alleles carries an upstream miniature inverted-repeat transposable element (MITE)-like insertion, which may be responsible for its increased expression in tissues with somatic embryogenesis. Antisense silencing of *CiRKD1* genes in transgenic tissues leads to a complete loss of somatic embryogenesis [114]. However, the presence of multiple *CiRKD1* gene copies and its location in a genomic region of about 80 kb including other 11 genes [121,126] may render embryogenesis activation by gene-editing approaches more complex.

Another gene controlling somatic embryogenesis is the *BABY BOOM* gene (*BBM*), which is able to induce embryos in microspores of *Brassica napus* and somatic cells of *A. thaliana* [91,127]. *BBM* genes belong to the AP2/ERF family of transcription factors encoding an AINTEGUMENTA-LIKE (AIL) APETALA2/ethylene responsive element-binding factor carrying two APETALA2 (AP2) DNA-binding domains [128], and have a crucial role regulating totipotency and embryonic identity [102].

Regulatory acquisition of embryogenic competence through genetic modification of a single gene (either *CiRKD1* or *BBM* genes) in somatic cells within the ovule is feasible. Yet, the challenge for mimicking sporophytic apomixis is induction of embryogenesis in somatic cells while simultaneously postponing zygotic development in the gametophytic tissue. Using genes acting in the post-fertilization processes dependent on the paternal allele might be a plausible option to delay or arrest the progression of the fertilized egg-cell. In rice, paternal expression of *BBM1* and its paralogues is required to bypass the fertilization checkpoint and transit to zygotic embryogenesis [104]. However, the zygotic transition postfertilization is initiated by asymmetric activation of parental genomes, with most genes playing major roles in the early development of plant embryos being maternally expressed [129,130]. Yet, *de novo* post-fertilization epigenetic reprogramming and transcriptional silencing of the paternal genome is controlled by *DNMT3A* in mice, a DNA methyltransferase highly expressed in oocytes [131]. *Dnmt3a* maternal knockout embryos die during post-implantation development [131]. Thus, using a plant ortholog of *DNMT3A* might help to unblock paternal gene expression in the early embryo and possibly result in zygotic embryo lethality without affecting the development of somatic embryos.

### 5.2. Mimicking Gametophytic Apomixis 

As few as three genes may be needed to mimic gametophytic apomixis, provided individual genes can create the desired changes in each reproductive module. Under such a simple model, this means (1) producing unreduced non-recombinant gametes (either by skipping meiosis or by blocking the reductional division of meiosis), (2) developing an embryo parthenogenetically, and (3) promoting endosperm development to complete the formation of a seed.

For mimicking gametophytic apomixis, researchers have a wide collection of meiosis related mutants to work with (e.g., [103,132]), though most of these show feeble phenotypes and low expressivity (see discussion below).

While the formation of unreduced gametes might implicate changing the regulation of simple genes related to RNA-directed DNA methylation pathways inducing either aposporous-like (e.g., through *AGO9*, [66]) or diplosporous-like (through *AGO104*, *SWI1*, or *DMT* genes, [133,134,135]) ovule progression, other strategies like annulling the main features of the meiotic division would require changes in more genes. A number of mutants affecting specific steps of the meiotic prophase and both meiotic cell divisions can be combined to convert meiosis into a mitosis-like division. For example, a triple knockout of the meiotic genes *SPO11-1*, *REC8* and *OSD1* can be used to generate MiMe (Mitosis instead of Meiosis) *Arabidopsis* plants. The resulting meiotic mutant phenotypes eliminate DNA double-strand breaks, meiotic recombination and chromosomal pairing (*Atspo11-1*, [108]) and destabilize centromeric cohesion and thus modifies chromatid segregation by impeding monopolar orientation of the sister kinetochores at metaphase I (*Atrec8*, [136]). Finally, the second division is omitted, likely by modulating the anaphase promoting complex/cyclin levels at the end of first meiosis (*Atosd1*, [110]). 

Using a similar MiMe strategy in rice, Mieulet et al. [111] combined knockouts of *REC8* and *OSD1* genes with *PAIR1*, a gene controlling homologous chromosome pairing, and suggested another three genes to be used instead of *SPO11-1*. Like in *Arabidopsis*, the triple rice mutant *pair1*, *Osrec8*, and *Ososd1* produced unreduced diploid gametes. Induction of a certain phenotype between different plant species and the use of a particular gene editing technology will depend upon synteny, protein sequence similarity and interactions with regulatory networks. For example, *OSD1* orthologs are single genes in *Hordeum vulgare* and *Brachypodium distachyon*, but are tandem duplications in *Zea mays*, *Sorghum bicolor*, and *Setaria italica* [137]. Thus, exploitation of such a gene for induction of unreduced gametes will require the use of RNAi or additional disruption of redundant gene copies to deal with duplications.

The next step toward clonal offspring needs to either skip fertilization or the incorporation of male chromosomes into the diploid female gamete. This has been tested using haploid induction genes that promote gynogenesis, i.e., the development of a fertilized egg-cell carrying only maternal chromosomes. One of these genes is a patatin-like phospholipase A restricted to the pollen tube, which might cause sperm chromosome fragmentation and paternal genome elimination in the fertilized egg cell [116]. The gene was characterized almost simultaneously by three research groups and called *MATRILINEAL* (*MTL*; [112]), *NOT LIKE DAD* (*NLD*; [115]), and *ZmPLA1* [116]. Haploid induction can be induced in rice by knocking out the gene *OsMATL* [117], and *TaMTL* in wheat [138]. Combining either the MiMe strategy with *MATL* successfully produced clonal seed, but at exceptionally low frequencies (Table 2; [105,139]). Similarly, adding a modified CENH3 to the MiMe or *dyad* phenotypes also eliminates the paternal genome postfertilization, creating clonal offspring in *Arabidopsis*, but at exceedingly low levels (Table 2; [140]). Alternatively, formation of unreduced gametes can be combined with genetic modification of *BBM*/*BBM*-like genes. The ectopic expression of these genes in egg cells before fertilization induces parthenogenetic embryos in pearl millet, maize, and rice, though not in *Arabidopsis*, at variable but overall low rates [67,104,141]. Loss-of-function mutations, such as those used for haploid induction, or spatial-temporal regulatory changes can be achieved by introducing indels in coding regions [112,115] or carrying out gene and promoter swaps with the CRISPR/Cas system. However, even when unreduced gametes can be produced at almost wild-type levels, apparent ineffective molecular coordination between distinct reproductive modules (i.e., sporogenesis and gametogenesis to form unreduced gametes, egg-cell parthenogenesis, and endosperm progression to produce a seed) fail to deliver plants with high expressivity of synthetic apomixis, with plants showing drastically reduced fertility in all cases (Table 2). The induction of synthetic apomixis, therefore, remains limited by low efficiency. Although the search for ‘silver bullet’ inducer genes seems momentarily appropriate, exploiting *de novo* apomixis will rely more on understanding the molecular interaction and background responsible for the low expressivity of combined targeted modifications.

The last step to form clonal seeds is the development of the endosperm. Even though combining genetically modified genes is strictly necessary to produce unreduced gametes and parthenogenetic embryos, inducing the formation of the endosperm without fertilization is not crucial for developing synthetic apomixis. Although fertilization in MiMe + *BBM1* rice plants limits the formation of clonal offspring to 29%, with most of the unreduced gametes forming tetraploid progeny [104], a proportion of such clonal seed discounting can be reduced in alternative ways. For instance, by targeting polyspermy avoidance mechanisms [142] or mechanisms needed for gamete fusion (such as secreting EC1 protein and further translocation of sperm specific gamete fusion proteins to the egg-cell surface; [143]). Even when the ratio between the endosperm and the embryo shifted from the normal 3:2 to 3:1 after fertilization of the unreduced central cell in MiMe + *BBM1* plants [104], the 2:1 maternal-to-paternal genome ratio in the endosperm required for appropriate development was maintained together with formation of viable seeds.

Engineering the autonomous formation of the endosperm without fertilization to obtain a complete asexual system may be ideal from a biotechnological perspective. From a biological viewpoint, despite studies showing that a single dominant locus is able to induce the autonomous endosperm phenotype in apomictic *Hieracium* [144], inducing autonomous endosperm in sexuals might be hard due to the molecular complexity of its development. Autonomous endosperm development differs between dicots and monocots, and the dynamics underlying this complex process rely on genome balance, epigenetic gene regulation, and parent-of-origin effects founded upon the contribution of the male gamete [145]. In addition, still unknown genetic modifier elements, protein interactions, and regulatory pathways underlying embryo-endosperm developments add to the complexity [101,146,147] and restrain autonomous endosperm expressivity [144]. Regardless of mutations in *fertilization independent endosperm* (*FIE*) and other Polycomb group genes leading to autonomous endosperm development in *Arabidopsis* [148], orthologs reported in both rice (*OsFIE1* and *OsFIE2*) and maize (*ZmFIE1* and *ZmFIE2*; [149]) produce distinct phenotypes. However, most plants (including crops) have hermaphrodite flowers and pseudogamous development of the endosperm via fertilization is possible in natural and synthetic apomictic plants. Considering, in addition, the lack of specific genes to be targeted for autonomous endosperm development, this trait is not essential per se for inducing apomixis.

### 5.3. Tuning Changes for Complete Penetrance, and High Expressivity and Fertility

Penetrance measures the percentage of individuals in a population who carry a specific gene (genotype) and express the gene-related trait (phenotype). Penetrance is complete (or incomplete) when all (or less than 100%) of the individuals with a specific genotype express the corresponding phenotype. The degree of expression of a trait is called expressivity and is used to describe variation among individuals with a specific gene (genotype) [150]. In the case of plant reproductive traits, and apomixis specifically, any apomictic individual showing the apomictic phenotype will contribute to the penetrance of the apomixis trait in the population, and any individual variation of the proportions of apomictic or sexual flowers will determine the expressivity of the trait on that particular genotype. While the penetrance of apomixis in natural populations is typically 1 (no natural sexuals coexist with apomicts), observed expressivity of apomixis in individual plants is generally high (expressivity = 0.89, Figure 3). In laboratory plants modified to express certain phenotypes, expressivity rather than penetrance can be determined. Plants genetically modified to produce clonal gametes usually exhibit high expressivity, while plants modified for clonal seed formation exhibit low expressivity of the trait (Table 2). The cause for the apparent uncoupling between reproductive modules for unreduced gametes and parthenogenesis in synthetic apomicts remains to be resolved. An important role in the uncoupling is likely played by interaction between modified genes and gene networks. Since MiMe phenotypes display high expressivity, the low expressivity for clonal seeds and the occurrence of higher ploidy progeny (see e.g., Khanday et al. [104]) can likely be attributed either to genes for parthenogenesis having secondary roles in the expression of the phenotype itself, or, most likely, on pleiotropic effects and genetic redundancy in the development of the phenotype.

**Figure 3 genes-11-00781-f003:**
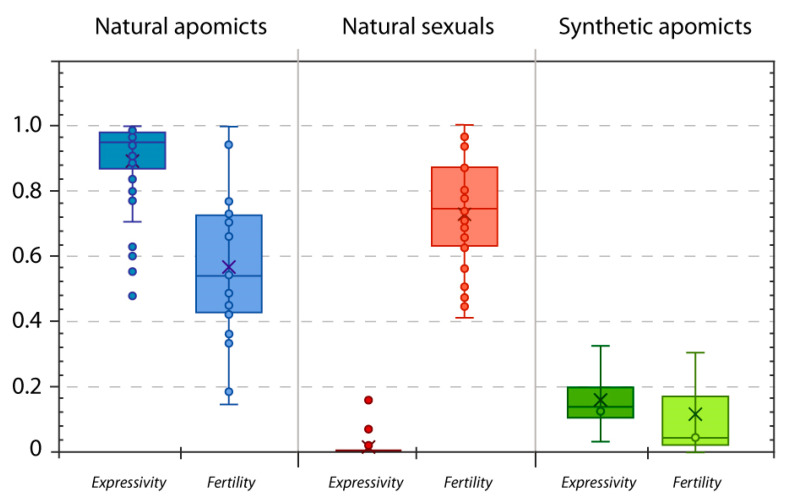
Box and whiskers plot representing observed variability in expressivity and fertility of apomixis in different biological conditions. In natural apomictic genotypes, apomixis expressivity is high (*n* = 35; average = 0.89 ± 0.14, max = 1, min = 0.48) and exhibits medium fertility *(n* = 18; average = 0.57 ± 0.24, max = 1, min = 0,15). In natural sexuals, only traces of apomixis have been recorded in specific cases (*n* = 33; average = 0.007 ± 0.027, max = 0.15, min = 0.0) and these plants exhibit high fertility (*n* = 35; 0.73 ± 0.17, max = 1, min = 0.41). Synthetic apomicts created so far show very low expressivity (*n* = 4; average = 0.16 ± 0.012, max = 0.33, min = 0.03; Table 2) and very low fertility (*n* = 3; average = 0.12 ± 0.16, max = 0.3, min = 0.002; Table 2). The graph is based on data from 42 species showing diplosporous and aposporous apomixis, autonomous and pseudogamous seed development, and their sexual conspecific taxa. Because gametophyte frequencies underestimate apomixis expressivity, all data were collected from studies on flow cytometry on seeds developed under open pollination (except for *Erigeron* in which parthenogenetic development was confirmed through embryology; [151]). Seeds developed through syngamy of meiotic gametes or meiotic plus apomeiotic gametes (i.e., incomplete apomixis and B_III_ hybrids) were classified as sexual. In species where data from multiple genotypes were available, only the mean value was used to avoid sampling bias (e.g., in *Paspalum intermedium*, *n* = 18 genotypes, expressivity = 0.86 ± 0.08, fertility = 0.19 ± 0.11; [152]). In some cases, fertility data were missing because of technical constraints (e.g., *Hypericum* forms several thousand seeds *per* fruit). Species included belong to the genus *Boechera* (*n* = 8, [153,154,155]), *Boehmeria* (*n* = 1, [156]), *Erigeron* (*n* = 2, [151]), *Hieracium* (*n* = 10, [157,158]), *Hypericum* (*n* = 1, [159]), *Paspalum* (*n* = 9, [152,160,161,162,163,164,165]), *Pilosella* (*n* = 2, [166]), *Poa* (*n* = 3, [167]), *Ranunculus* (*n* = 4, [168,169]), *Taraxacum* (*n* = 2, [170]).

To some extent, gene editing can help tune and increase the levels of parthenogenesis and formation of clonal seeds. For instance, after a comprehensive genetic analysis of haploid induction in maize [171], distinct studies identified the *MTL*/*ZmPLA1*/*NLD* gene and took advantage of gene editing to obtain plants with 6.7% haploid induction rate [112], or to combine *mtl*/*zmpla1*/*nld* with a *zmdmp* mutant to increase the haploid induction rate 5–6-fold [172]. These works were a step forward to accelerating crop breeding through in vivo haploid induction systems. There is now a strong opportunity to search for new genes and explore alternative combinations of gene edits to create MiMe-like phenotypes that better interact with parthenogenetic genes and vice versa. A number of meiotic related genes are known to play relevant functions in specific steps, e.g., for recombination initiation [173]. In addition, various genes responsible for acquisition of an embryogenic cell fate [101,102] are available for being tested in different species and under distinct gene-edited configurations. Examples include the *LEAFY COTYLEDON* (*LEC*) family of transcription factors, the *SOMATIC EMBRYOGENESIS RECEPTOR KINASE* (*SERK*), *WUSCHEL* and *AGAMOUS*-*Like 15* (*AGL15*), *WOUND INDUCED DEDIFFERENTIATION 1* (*WIND1*), and the homeobox gene *BELL1*, among others [174,175,176,177,178].

For efficient expression of an apomixis-like pathway in sexual plants, researchers will need to bring to bear gene expression modification, precise base editing and trait optimization using allelic series on different gene targets to increase trait expressivity. Single genes can be knocked out in different ways, creating specific changes in metabolic pathways and cellular phenotypes. Additionally, different plant mutants can produce similar phenotypes (e.g., multiple MMC or embryo sacs). Therefore, we expect future *de novo* engineering of apomixis to involve appropriately modifying different genes to mimic apomixis-like steps, modulating their expression, and optimizing the fittest combination of mutants toward full trait expressivity. All natural apomicts display complete penetrance and high expressivity (Figure 3). Increasing *de novo* apomixis efficiency will require expanding data collection (see Section 6) and compatibility among apomictic plant systems to provide useful information about the involved metabolic pathways, gene networks and protein interactions.

Another significant reproductive trait in genetically modified plants is fertility, i.e., the proportion of seeds formed in comparison to the total number of developing ovules. While in studies on mutants affecting distinct plant traits, fertility might not be that important, in the present case it is of special interest because (1) we are modifying reproductive development, and such changes will impact the total amount of developing ovules, and (2) inducing phenotypic changes associated with negative effects on plant fertility is undesirable or in opposition to plant breeding goals. Despite the obvious differences between estimates of the proportion of clonal seeds and fertility, in some studies on plants modified for clonal seed formation, there is a lack of clear information about fertility (Table 2). As an example, a plant might produce a total of 10 viable seeds, five being clonal, from a total of 100 developing ovules. If fertility is not considered there might be a misreading of the expressivity of the trait (as 50% instead of the real 5%) in that plant. Here as well, apomicts display higher fertility compared to synthetic apomixis mutants, but lower compared to sexual conspecific taxa (Figure 3). A reduction in fertility is inherently associated with apomixis, and hence, it is a crucial attribute to consider in any attempt to induce apomixis. Increasing data collections about fertility in apomicts, sexuals and mutants will be necessary for understanding changes in fertility caused by modifications of developmental programs and plant reproductive modules, including those observed in natural apomicts and genetically modified plants.

Experimental studies on cultivar developments and wild species have shown apomixis is a dominant trait and is transgenerationally stable [10,179]. Yet, unstable phenotypes occur by spontaneous uncoupling of reproductive modules and formation of segregating sexual offspring (either B_II_ or B_III_). Phenotypic instability due to gene-environment interactions, genetic heterogeneity or genetic compensation effects has also been observed in mutants [124,180]. Even if complete penetrance of a synthetic phenotype is attained in a natural population or a crop field, trait expressivity will likely vary due to genotype-by-environment interaction. In natural apomicts, experimental assays demonstrated an influence of climatic factors such as daylength, temperature, and salinity on proportions of apomixis and sex [152,181,182,183,184]. Environmental influence on sexuality and apomixis was recently exposed in situ on natural populations of facultative apomictic *Paspalum intermedium* [152]. In apomictic grass cultivars, differential rates of residual sexuality may cause loss of productivity due to segregating offspring, impacting cultivar management strategies [163].

The above aspects influencing penetrance, expressivity and fertility should be rigorously screened and evaluated while establishing a synthetic apomictic crop. For wider market release, the aim will be to produce a fertile highly expressive apomict, with no off-target effects of gene editing.

## 6. The Remaining Challenge of Data Collection for Genomic Dissection of Apomixis Loci

Most of the over 600 plant reference genomes publicly available via GenBank represent model or crop plants and their close wild relatives. Collections of genomic data from apomictic plants are still meagre, and no apomictic plant genome has yet been assembled. By comparing the list of species with sequenced genomes to those genera in which apomixis is present in at least one species [185], we found that a total of 60 species that belong to 33 apomictic-containing genera have a GenBank genome assembly (Appendix A). The sequenced species are sexual, mostly diploids. The majority of such species belong to genera containing species with sporophytic apomixis (39 species), with *Citrus* being the most common genus (nine species). Genome assemblies are also available for species belonging to intensively studied apomictic genera, such as *Boechera*, *Boehmeria*, *Cenchrus*, *Eragrostis*, *Erigeron*, *Hypericum*, *Panicum*, and *Setaria* (13 species). Hence, the currently available data represents an interesting collection of sexual species having conspecific apomicts. These data will likely mark a direction and foster future research on apomixis.

We also found another 54 species belonging to six genera in which sporadic occurrence of apomixis or elements of apomixis had been reported (including *Solanum* and *Oryza*, having 20 and 16 species and varieties sequenced, respectively). In these cases, some reports on apomixis related features date back to the 1950s or earlier (e.g., [186,187]), and reported attributes are likely to be genotype-dependent and thus, unless alike genotypes are analyzed, the information about apomixis that could be extracted from these genomes is probably limited. 

Although no apomictic genomes are yet available, the dropping cost of genome sequencing is enabling sequencing projects with broad phylogenetic sampling across thousands of non-model plants [188]. Undoubtedly, apomictic genomes will become available, and these genomes can be mined together with those of conspecific sexuals as an invaluable resource to uncover the basis of natural apomixis in plants. Comparative whole genome analysis can detect apomixis-associated regions such as the ASGR locus in *Pennisetum squamulatum* [67,189] or the ACL in *Paspalum simplex* [89,190] and help better understand how sequences related to apomixis are co-evolving within and among plant lineages. A further example of apomixis-related comparative genomics is the discovery of the apomixis candidate gene *CitRKD1*, which relied on *de novo* assembly and sequencing of cultivated *Citrus* and wild relatives [121]. Reference genomes are also useful to align population-level sequencing data and identify genetic differentiation between sexual and apomictic individuals within a species. At broader phylogenetic scales, trends in the evolution of apomictic loci can emerge that pinpoint common apomixis-associated genes, allowing these to be targeted in diverse crop species. Although genome editing is not limited to introducing naturally occurring variation, comparative genomic data can help identify and prioritize editing targets.

Before high-quality apomictic plant genomes become widely available, several technical challenges in their assembly will need to be overcome. *De novo* whole-genome assembly of plants is generally challenging due to large genome sizes, variable ploidy and extensive duplications, repetitive elements, and areas of high GC or AT content. In contrast to most crops, wild species can also harbor high levels of heterozygosity, complicating genome assembly. As mentioned above, apomicts are frequently polyploid and may exhibit apomixis-associated regions with high repetitive element content [88,99,191]. These issues can impede accurate assembly of apomixis-associated genomic regions, especially when relying on short reads. Genomes assembled from short reads are generally draft genomes, comprised of thousands of contigs, with many gaps and errors. Unresolved haplotypes leave us without access to a whole layer of genetic variation information. Long-read sequencing, linked-read strategies, and optical mapping offer significant improvements to genome assembly quality [192], though often at substantially higher costs. However, even when accurate and phased assemblies exist, a single linear reference genome only provides a limited view of within-species sequence diversity. As the apomixis trait can be hemizygous and may thus involve complex structural variation, reference bias could undermine analyses of the trait. One approach to address this issue is by constructing plant pangenomes based on population-level genomic sampling [193]. The pangenome reveals genes with presence/absence variation and can lead to trait discoveries such as the recent tomato pangenome analysis that identified the *TomLoxC* gene as an important player in fruit flavor [194]. Alternatively, genomic sequencing reads can be broken down into k-mers, which can then be associated with phenotypes without requiring computationally expensive and error-prone assembly [195].

Building on a layer of genomic information also allows the integration of functional data on transcripts and proteins to better infer interactions between genes in networks. Substantial advances in this field include analysis of RNA-seq and open chromatin data from bulked and single cells [196,197,198] as well as in silico methods to infer functions such as binding properties from amino acid sequences [199]. Improving our knowledge on how genes are interconnected in different metabolic pathways will be important for editing different reproductive genes without disrupting the processes that lead to formation of a mature seed. Apomixis-associated genes may interact with other genes requiring multigene editing to alter a trait, as illustrated by genetic redundancy and/or cross-regulation among the five RKD transcription factors that regulate egg cell differentiation [123]. A lineage-specific view of gene regulation will be important when studying regulatory pathways in apomicts because these shows considerable differentiation. Homologous genes can show different regulatory expressions. For instance, CHR106/DDM1 was downregulated in *Tripsacum dactyloides* apomictic plants [66], but showed no difference in *Boechera holboellii* apomictic and sexual plants, and was reported as upregulated in *Eragrostis curvula* [200]. 

Substantial challenges may lie ahead on the path to induction of penetrant apomixis, but recent progress is encouraging. The more integrated view of genetics enabled by new technologies and data makes it an ideal time to investigate complex traits such as apomixis. A better understanding of the molecular basis of apomixis through exploitation of next-generation sequencing tools in different types of apomicts while guaranteeing data compatibility among studies [35] will be central to implementing gene editing tools in the creation of a fertile synthetic apomict displaying high expressivity. Although most possible alleles can be generated using gene editing, natural allelic variation in apomixis-associated loci can be used to narrow the choice of target allele. Nature has tinkered with alleles over evolutionary time, and we can harness the results rather than reinventing the wheel entirely with editing.

## Figures and Tables

**Figure 1 genes-11-00781-f001:**
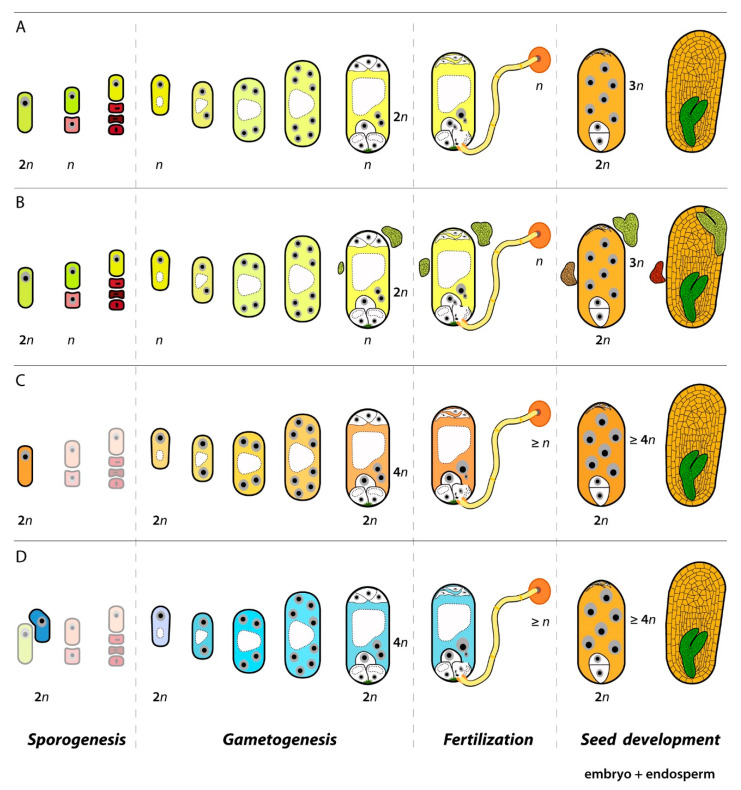
Pathways of seed formation in plants. At least one or two reproductive modules or developmental programs are distinctively altered between sexual (**A**) and apomictic developments (**B**–**D**). In sporophytic apomixis, a clonal embryo developed from the surrounding somatic tissue replace the zygotic embryo (**B**). In gametophytic apomixis, sporogenesis, or the acquisition of a megaspore mother cell identity and progression through meiosis is the first altered developmental program. In diplospory (**C**) meiosis is bypassed, while in apospory (**D**) it is highly depleted and surrounding somatic cells acquire a megaspore-like fate. Gametogenesis, or the formation of the female gametophyte, is generally conserved in the three pathways. The second developmental step that is altered is fertilization, i.e., the delivery of male gametes and fusion with the female ones. While in sexual plants double fertilization happens (**A**), in most apomicts only one fertilization event is possible due to parthenogenesis and this fertilization is required for developing the endosperm (**C**,**D**).

**Table 1 genes-11-00781-t001:** Common genes used to manipulate specific reproductive steps during gamete and seed formation.

Gene	Function	Mutant Phenotype	Reproductive Module	Expressivity	Reference
*SWI1*/*dyad*	sister chromatid cohesion	Arrested meiosis	Meiosis	0.99 ^1^	[106]
*OsPAIR*	Homologous chromosome pairing	Arrested meiosis	Meiosis	1.00	[107]
*AtSPO11*-*1*	DSBs and initiation of homologous chromosome recombination	Lack recombination	Meiosis	0.97 ^2^	[108]
*AtREC8*	sister chromatid cohesion	Univalents; aberrant chromosomal segregation	Meiosis	1.00 ^3^	[109]
*AtOSD1*	Entry into MII	Lack meiosis II; dyad formation	Meiosis	0.85 ^4^	[110,111]
*ZmMATL^5^*	Sperm-specific phospholipase	Haploid induction, haploid seeds	Fertilization	0.07 ^6^	[112]
*AtCENH3*	Centromere-mediated genome elimination	Haploid seeds	Embryogenesis	0.08 ^7^	[113]
*RKD*	Transcription factor	Somatic embryogenesis	Embryogenesis	1 ^8^	[114]
*BBM*	Transcription factor; embryo development	Somatic embryogenesis	Embryogenesis	! ^9^	[91]

^1^ in a very low proportion (0.24%), one dyad cell can develop into a mature embryo sac and produce a seed [106]; ^2^ mutants are semi-sterile with females producing 3% mature female gametophytes and three seeds per silique when female mutants were used in crosses to the wild type ecotype (wild type seed set per fruit = 45 ± 5; [108]; ^3^ mutants show complete male and female sterility [109]; ^4^ dyads in *osd1* mutants produce on average 35 ± 6 polyploid (3× and 4×) seeds per fruit compared to 38 ± 11 of the wild type [110]; ^5^ also named *NLD* [115] and *ZmPLA1* [116]; ^6^ haploid induction rates of 4–12.5% (average 6.65%) in maize; haploid induction rates of 2–6% in rice [117], seed-setting rate in rice 11.5% [105]; ^7^ estimated from circa 12% fertile ovules multiplied 62.5% of haploid seeds; crosses between GFP-tailwasp X wild-type plants produced 25–45% viable haploid offspring (the rest corresponded to diploids and aneuploids); no clear information on ovule abortion or seed-set is provided; ^8^ loss of *RKD1* function by antisense overexpression abolished somatic embryogenesis in transgenic *Citrus* and the transgenic T1 plants were derived from self-pollinated zygotic embryos [114]; ^9^ overexpression using semiconstitutive promoters induces ectopic embryo formation in leaf tissues and other pleiotropic effects; thus, there are no reproductive units per se, and neither penetrance nor expressivity can be estimated as in the other cases.

**Table 2 genes-11-00781-t002:** Combination of mutants used to create unreduced nonrecombinant gametes and clonal progeny.

Gene Combination	Reproductive Phenotype	Expressivity	Fertility ^1^	Reference
*AtSPO11*-*1* + *AtREC8* + *AtOSD1*	Unreduced nonrecombinant gametes	1.00	0.66 ^2^	[110]
*OsPAIR1* + *OsREC8* + *OsOSD1*	Unreduced nonrecombinant gametes	1.00	0.74 ^3^	[111]
*AtSPO11*-*1* + *AtREC8* + *AtOSD1* + *GEM* ^4^	Clonal offspring (mixed) ^5^	0.33	0.3 ^6^	[140]
*dyad* + *GEM* ^4^	Clonal offspring (mixed) ^5^	0.13	0.0018 ^7^	[140]
*AtSPO11*-*1* + *AtREC8* + *AtOSD1* + *BBM1*	Clonal offspring (mixed) ^5^	0.11–0.29	? ^8^	[104]
*OsPAIR1* + *OsREC8* + *OsOSD1* + *OsMATL*	Clonal offspring (mixed) ^5^	0.02–0.04	0.045	[105]
*OsSPO11*-*1* + *OsREC8* + *OsOSD1* + *OsMATL*	Clonal offspring	? ^9^	? ^9^	[139]

^1^ fertility is considered as a seed set or number of seeds formed from the total number of ovules; the data were collected from each study without considering germinability (which ranged between 73–92% among studies). When the number of ovules per fruit were not provided, the number of seeds were compared to those of the wild type plant; ^2^ dyads in this triple mutant produce on average 25 ± 6 polyploid (3× and 4×) seeds per silique compared to 38 ± 11 of the wild type [110]. MiMe rice plants produce 81.2% of seeds (all 4x derived from unreduced gametes) compared to 79.2% in the wild type [105]; ^3^ from a total of 1012 seeds from 1370 ovules (flowers) analyzed [111]; ^4^ GEM line called for Genome Elimination caused by a Mix (GEM) of CENH3 variants (Marimuthu et al. [140], supporting online material page 5); ^5^ clonal offspring were intermixed with polyploid and aneuploid offspring; ^6^ considering fertility as 15 seeds per silique in comparison to 50 seeds per silique in the wild type (Marimuthu et al. [140], supporting online material pages 12–13); ^7^ considering fertility as 0.9 seeds per silique in comparison to 50 seeds per silique in the wild type (Marimuthu et al. [140], supporting online material pages 12–13); ^8^ no proportion of seed set in comparison to total number of ovules/flowers is mentioned; ^9^ Xie et al. [139] provide cytological evidence of MiMe induction in a gene-edited rice plant, but no data about formation of clonal progeny by the edited *OsMATL* gene, nor about fertility of the gene modified plant.

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
