# Peer review of "Can We Use Gene-Editing to Induce Apomixis in Sexual Plants?"

_genes, 2020, doi:10.3390/genes11070781_

Round 1

Reviewer 1 Report

The ability to quickly switch the type of reproduction of cultivated plants from sexual to apomictic would give the breeders a powerful tool and would allow them to fix economically useful traits in stable heterozygous lines that are now appearing only in the first generation. Apomixis is widespread in plants and occurs in very distant taxa, which indicates its repeated origin, affecting various elements that control the reproduction system. Unfortunately, these control elements are still poorly studied and our information concerns only a small number of objects. Therefore, the review and discussion work proposed by the authors is of considerable interest.
The authors discuss in detail the mechanisms of apomictic reproduction existing in nature and the possible ways of their evolutionary origin. The paper considers a number of genes and gene complexes involved in the regulation of the reproduction system, changes of which can lead to apomixis. The possibility of modifying these genes using various methods of precise gene targeting is being considered.
The authors propose to enter into a discussion about the possible ways of creating artificial apomixis and to investigate the fine molecular mechanisms of its possible induction. The work is of considerable interest to all plant molecular biologists.

Author Response

Thank you for your evaluation.

Reviewer 2 Report

I enjoyed reading this review. The scientific topics are interesting and up to date, with very nice drawings. It may be even useful to students.

I could not find any particular flaw and i think it will be considered of interest by many readers. The only very small flaw is maybe a low interest in the mechanisms of megaspores formation. Only as an example to take a look at the reference therein i remember this article "Papini A., Mosti S., Milocani Eva, Tani Gabriele, Pietro Di Falco and Luigi Brighigna. (2011) Megasporogenesis and Programmed Cell Death in Tillandsia (Bromeliaceae). Protoplasma, 248: 651-662."

However, the review is alredy of good quality in the present form.

Author Response

Thank you for your observations.

We are aware that there is some diversity in the patterns of megaspore formation, specially in polysporous and diplosporous species. However, considering these details was a bit out of the scope of the present discussion and the manuscript already had a considerable length to include more specific issues about the development of apomixis.